# Dynamic Characteristics of Urbanization Based on Nighttime Light Data in China’s “Plain–Mountain Transition Zone”

**DOI:** 10.3390/ijerph19159230

**Published:** 2022-07-28

**Authors:** Tingting Li, Zengzhang Guo, Chao Ma

**Affiliations:** 1School of Surveying and Land Information Engineering, Henan Polytechnic University, Jiaozuo 454003, China; 111804010002@home.hpu.edu.cn (T.L.); gzc@hpu.edu.cn (Z.G.); 2Department of Surveying and Mapping Engineering, Henan College of Surveying and Mapping, Zhengzhou 451464, China; 3Key Laboratory of Spatiotemporal Information and Ecological Restoration of Mines (MNR), Henan Polytechnic University, Jiaozuo 454003, China; 4Research Centre of Arable Land Protection and Urban–Rural High–Quality Development in Yellow River Basin, Henan Polytechnic University, Jiaozuo 454003, China

**Keywords:** prolonged artificial nighttime-light dataset of China (PANDA), China’s “plain–mountain transition zone”, nighttime lighting types, urbanization, geomorphology

## Abstract

China’s “plain–mountain transition zone” (hereinafter referred to as the “transition zone”) has experienced rapid and diverse urbanization processes. Assessing the dynamic characteristics of urbanization is particularly important for sustainable development of the transition zone. Nighttime light (NTL) data have been widely used to monitor urbanization. Based on the prolonged artificial nighttime-light dataset of China (PANDA) from 1984 to 2020, we partitioned the nighttime light of the study area into four types (low, medium, high, and extremely high) by adjusting the threshold of the brightness gradient (BG) method. The spatiotemporal characteristics of urbanization in 426 districts and counties of 71 prefecture-level cities in the transition zone were analyzed. Our results indicated that the middle region of the transition zone (Yanshan Mountains and Taihang Mountains) experienced the fastest urbanization development, and the urban expansion speed broke through the topographic limitation of the plain–mountain. However, the rapid development of urbanization in the middle plains resulted in the nighttime lighting area (NTLA) tending to become saturated, which caused an unsustainable potential crisis in urban development in this area. Urbanization was mainly manifested in the transition of the low nighttime lighting type (NTLT) to the medium NTLT or higher NTLT. The northern region of the transition zone (Greater Khingan Mountains) experienced the slowest urbanization development, with the lowest nighttime lighting density (NTLD) in the northern mountainous area, where the urbanization was mainly manifested by the expansion of the low NTLT. The urbanization development of the southern region in the transition zone (Wushan and Xuefeng Mountains) was at a medium level, and the urbanization of the plain in the southern region was also better than that of the mountainous area. Urbanization was mainly manifested in the expansion of the low NTLT, supplemented by the transition from the low NTLT to high NTLT. Whether in the north, middle, or south of the transition zone, the plain–mountain topographic variations caused a gap in urbanization, making the urbanization development of the mountains and plains unbalanced.

## 1. Introduction

Urbanization has become the main form of economic development in developing countries [1]. As the main form of urban and rural development, urban dynamic changes not only reflect the relationship between urban and rural areas, but also comprehensively reflect the development stage of urban and rural areas [2]. Urban dynamic change is a multidimensional concept including the population, economy, and geographical space, which is often accompanied by the growth of the urban population, the intensification of social and economic activities, the expansion of the urban construction scale, and so on [1,2]. As a common phenomenon in the growth of developing countries, urbanization also causes social problems such as air pollution, resource depletion, and environmental fragility, which brings major challenges to sustainable development [3]. The ecological environment is affected by urbanization, and studying the dynamic characteristics of urbanization can help to understand the urban development better and predict the changing trends of urbanization in the future [4]. Analyzing the impacts or threats of urban dynamic changes is of great significance to guide the protection of ecological resources, ensure regional sustainable development, and promote the harmonious coexistence between man and nature [5,6,7,8,9].

Remote sensing can avoid dependence on the system and manpower and is one of the important means to study global change [10]. A number of studies shows that nighttime light (NTL) is closely related to human socioeconomic space activities. At present, the urbanization research on the macro-regional scale has changed from traditional statistical analysis to spatiotemporal analysis represented by NTL data [11]. For example, Henderson et al. [12] used the threshold method to extract urban areas in San Francisco, Beijing, and Lhasa based on the Defense Meteorological Satellite Program-Operational Linescane System (DMSP–OLS) NTL data. Cao et al. [13] employed a region growth algorithm of the support vector machine (SVM) to semi-automatically extract urban areas from DMSP-OLS and SPOT Normalized Difference Vegetation Index (NDVI) data. Su et al. [14] extracted the built-up areas of eight cities in the Pearl River Delta based on DMSP-OLS NTL data using the neighborhood statistical analysis (NSA) method. Based on the DMSP-OLS data from 1992 to 2008, Liu et al. [15] used the local optimization threshold method to study the urban expansion dynamics of Chinese cities. Ma et al. [16] divided the NTL of 274 cities in China from 1992 to 2012 into five types using the brightness gradient (BG) method for DMSP-OLS data, and studied the development models of each city. Based on the DMSP-OLS data from 1992 to 2013, Jiang et al. [4] used the BG method to divide the NTL of more than 10 major countries in Africa into three types to study the urban dynamic changes of these countries. Based on DMSP-OLS data from 1992 to 2013, Kamarajuged et al. [17] used the BG method to divide the NTL of 15 major cities in Southeast Asia into three types to evaluate the urban growth dynamics. Zhao et al. [18] increased the research time series by matching the spatial resolutions and intra-year DN values of DMSP-OLS data and the National Polar-orbiting Partnership Visible Infrared Imaging Radiometer Suite (NPP–VIIIRS) data, and developed a new framework based on the spatial variation of the NTL gradient (SVNG) to map urban dynamics in Southeast Asia using consistent NTL data (1992–2018). Based on DMSP-OLS and NPP-VIIIRS (1993–2017), Li et al. [19] used the reference data-assisted comparison method to extract the built-up area of urban agglomerations and studied the temporal and spatial expansion of urban agglomerations around the Dongting Lake Ecological Economic Zone.

Through literature research, it was found that the commonly used NTL data are DMSP-OLS and NPP-VIIIRS data. DMSP-OLS data have the advantages of wide coverage and continuous acquisition time but have the disadvantages of poor timeliness (1992–2013) and oversaturation of radiated digital number (DN) values [20]. NPP-VIIIRS data have the advantages of high spatial resolution and strong radiation resolution but the disadvantages of a short time sequence (2012–2018) and background noise [21]. Therefore, when analyzing long-time series data, some researchers take the coincident year (2012) of the above two datasets as the corrected base year and then used it after matching the spatial resolution and DN value [18,19,22,23]. No scholars have published a standard dataset after correcting these two types of data, and scholars may obtain different correction results depending on the study area and research objectives. Based on NTL data, researchers proposed a variety of methods such as thresholding [12,24], image classification [13], and neighborhood statistical analysis [14] to analyze the urbanization, among which thresholding is the most commonly used. Thresholding can be subdivided into empirical thresholding (ET) [25,26], local optimization thresholding (LOT) [15,27], and brightness gradient-based thresholding (BG) [4,16,17]. Of these methods, ET is the simplest but does not have universal applicability, while LOT is more accurate but can only be used at a local scale. Compared to other thresholding methods, BG classifies NTL images according to the quadratic polynomial relationship between the DN value and the corresponding gradient brightness of each pixel, which can better reflect the temporal and spatial dynamic changes of the city, and has the characteristics of universal applicability and large-scale use [4,16,17]. Previous studies on the dynamic characteristics of urbanization mostly focused on individual cities or regions, but there has been no study on the large-scale geographical transition zone [28] and its response to topography. The geomorphic pattern of China’s three major ladders strongly shapes the national multi-level and multi-scale transitional geographical space, highlighting spatial heterogeneity, structural complexity, and functional diversity. Therefore, China’s second and third ladders are characterized by differences, changes, and uncertainties in the interaction of natural and human factors, and present the characteristics of large geographic transition zones. We propose three innovations for the problems existing in data types, research methods, and research areas: (1) A Prolonged Artificial Nighttime-light Dataset of China (PANDA) [29] was selected instead of DMSP/OLS and NPP/VIIRS data for the time period from 1984 to 2020, which is characterized by strong timeliness and long time series with high data quality. (2) We improved the BG method and adopted a unified classification standard to achieve large-scale urban nighttime light classification quickly. (3) China’s “plain–mountain transition zone” (hereinafter referred to as the “transition zone”) is not only the transition zone of China’s second and third ladders but also the economic transition zone from developed areas in the east to the less-developed areas in the west. This study area has the structural complexity caused by both natural and human factors, which is a new exploration different from the previous study area.

With the overall promotion of national development by reform and opening up, the problem of unbalanced urbanization development should be taken seriously [30,31]. At present, a comprehensive and continuous understanding of the uneven urbanization development, the spatiotemporal expansion characteristics of urbanization, and the response of urbanization to topography are still lacking in China’s transition zone. The primary purpose of this study was to assess the spatiotemporal characteristics of the urbanization dynamics from 1984 to 2020 in China’s transition zone by analyzing the spatial changes in the nighttime light dataset (PANDA) and to provide a scientific reference for urban coordinated development policies. First, the PANDA nighttime light dataset was resampled and cropped. Second, a quadratic relationship between the brightness gradient and DN value of each pixel was established, and the nighttime light of China’s transition zone was partitioned into four types (low, medium, high, and extremely high) by adjusting thresholding of the brightness gradient (BG) method. Third, the spatial and temporal characteristics of urbanization in 426 districts and counties of 71 prefecture-level cities in the transition zone were analyzed over 34 years. Finally, the correlation between the dynamic urbanization expansion and the topography of the transition zone was analyzed. Research on the dynamic characteristics of cities in the transition zone has important practical significance for understanding the development level and sustainability of urbanization, and has theoretical reference significance for formulating policies for the coordinated development of the eastern and western regions in China.

## 2. Study Areas and Data

### 2.1. Study Areas

China’s transition zone is a topographic change area formed by the Greater Khingan Mountains, Yanshan Mountains, Taihang Mountains, Wushan Mountains, and Xuefeng Mountains. The geomorphic type of the transition zone changes from the eastern plain less than 400 m to the central mountainous area more than 400 m in China. With reference to the definition method of the macroscopic geographic transition zone [28,32,33], this paper took the 400 m contour as the benchmark, took 80 km as the buffer zone firstly, and then took the overlapping area between the buffer area and the county-level administrative areas as the final boundary of the study area. Counties where the overlapping area of the county-level administrative area and the buffer area was less than 50% of the county area were discarded. The longitudinal range of the transition zone is 105°12′ E~127°32′ E; the latitude range is 21°42′ N~53°32′ N; and the azimuth is 32° (SSW-NNE) (Figure 1a). The transition zone starts from the Heilongjiang Province in the north, passes through the Jilin Province, Liaoning Province, the Inner Mongolia Autonomous Region, Beijing Municipality, Tianjin Municipality, Hebei Province, Shanxi Province, Henan Province, Hubei Province, Chongqing Municipality, Hunan Province, Guizhou Province, and reaches the Guangxi Zhuang Autonomous Region at the southern end, crossing 14 provincial-level administrative regions with an area of 11.36 × 10^4^ km^2^ (Figure 1b). The economic development on both sides of the transition zone is extremely unbalanced. The land area, population, and GDP in the East account for 10.91%, 56.66%, and 71.64% respectively, and the land area, population, and GDP in the west account for 89.09%, 43.34%, and 28.36%, respectively (based on the data of the China Statistical Yearbook in 2019) [34]. As the hub of China’s eastern and western regions, the transition zone is of practical significance for the coordinated development and sustainability of said regions to study the dynamic characteristics of urbanization.

### 2.2. Data Source

The Prolonged Artificial Nighttime-light Dataset of China (PANDA) from 1984 to 2020 was calculated by Zhang Lixian et al. [29] of Tsinghua University in 2021 using a NTL convolution long- and short-term memory neural network (ConvLSTM), released by the National Tibetan Plateau Data Center (http://data.tpdc.ac.cn/ (accessed on 21 October 2021)). The spatial resolution of this nighttime light dataset is 30 arc seconds (approximately 1 km), with a DN value ranging from zero to 63, which is a dimensionless quantity. The model evaluation between this dataset and the original image showed that the average root mean square error (RMSE) was 0.73; the coefficient of determination (R^2^) was 0.95; and the slope of pixels was 0.99, indicating that the data quality of the generated product was high. This dataset can well capture the time trend of newly built areas and can provide the potential NTL in the early stage, which can be used to describe the expansion types of different cities [29].

A digital elevation model (DEM) was jointly mapped by the National Aeronautics and Space Administration (NASA), National Imagery and Mapping Agency (NIMA), and the space agencies of Germany and Italy, completed by the Shuttle Radar Topography Mission (SRTM) (http://srtm.csi.cgiar.org/ (accessed on 30 March 2021)). The spatial resolution of the DEM is 90 m.

The administrative division data were obtained from China’s basic geographic information data at a scale of 1:4 million, based in 2018, provided by the National Basic Geographic Information Center (http://ngcc.sbsm.gov.cn/ (accessed on 20 September 2020)).

The land use dataset in China used was a remote sensing monitoring data product provided by the Resource and Environmental Sciences and Data Center (http://data.tpdc.ac.cn (accessed on 28 February 2022)) [35], which was generated by visual interpretation with Landsat TM/ETM remote sensing images as the main data source. The land use dataset with a spatial resolution of 1 km was classified into six classes: farmland, woodland, grassland, waters, construction land, and unused land.

The population data were obtained from the Socioeconomic Data and Applications Center (SEDAC) of NASA (https://www.worldpop.org (accessed on 5 March 2022)), with the unit of people per square kilometer and a spatial resolution of 30 arc seconds.

The gross national product (GDP) was obtained from the Resource and Environmental Sciences and Data Center (https://www.resdc.cn (accessed on 28 February 2022)), with the unit of million yuan per square kilometer and a spatial resolution of 30 arc seconds.

The characteristics and use of the data are listed in Table 1.

### 2.3. Data Processing

In the process of studying the urbanization in the transition zone, many types of data were used. The processes of data processing were as follows: First, to reduce the effects of geographic projection, the geographic projection of the used data was uniformly transformed to Lambert_China. Second, the DEM and administrative division data were used to extract the boundary of the transition zone. The extraction processing can be found in Section 2.1. Third, in order to compare the area between different data types, all raster images were resampled to a spatial resolution of 1 km. Fourth, the improved BG method was used to classify PANDA data for nighttime light, and the classification results were compared and verified using land use and population density data. Fifth, the spatiotemporal change trend and transitions of the nighttime lighting types in the transition zone were calculated, and the dynamic characteristics of urbanization were analyzed. Finally, the impact of the terrain on urbanization was analyzed, and the relationship between GDP, POP, and urbanization was discussed in the transition zone.

## 3. Methods

### 3.1. The Spatial Gradient of Nighttime Light

The Sobel operator can effectively suppress noise, weight the influence of the pixel position, and accurately locate the edge position. In order to describe the spatial changes for the NTL data, the Sobel operator was used to convolute the nighttime light images and detect the edge of the image (Figure 2).

The illustration in Figure 2 shows that NTL_5_ was the central grid cell, and the BG values were estimated by change rates in the horizontal (*dNTL/dx*) and vertical (*dNTL/dy*) directions (i.e., first-order derivative) from the central grid cell to its adjacent eight grid cells (*NTL*_1_, *NTL*_2_, *NTL*_3_, *NTL*_4_, *NTL*_6_, *NTL*_7_, *NTL*_8_, and *NTL*_9_) as follows:(1)$$BG=\sqrt{{\left(dNTL/dx\right)}^{2}+{\left(dNTL/dy\right)}^{2}}$$
(2)$$dNTL/dx=\left[\left(NTL_{3}+2NTL_{6}+NTL_{9}\right)-\left(NTL_{1}+2NTL_{4}+NTL_{7}\right)\right]/8$$
(3)$$dNTL/dy=\left[\left(NTL_{7}+2NTL_{8}+NTL_{9}\right)-\left(NTL_{1}+2NTL_{2}+NTL_{3}\right)\right]/8$$

### 3.2. The Relationship between Nighttime Light and the Brightness Gradient

By comparing the nighttime light with the brightness gradient (Figure 3), it can be found that the low value area of BG pixels mainly corresponds to the high and low value areas of NTL pixels, while the high value area of BG pixels corresponds to the medium value area of NTL pixels. Additionally, there is a quadratic relationship between NTL and BG values, as follows [16]:(4)$$BG=aNTL^{2}+bNTL+c$$
where *a*, *b*, and *c* are the coefficients of quadratic polynomials.

### 3.3. Spatial Partition of Nighttime Light Imagery

The brightness gradient-based thresholding (BG) method calculates pixel-by-pixel through the quantitative relationship between the NTL and BG values and divides the nighttime light imagery spatially. It does not need empirical thresholding and can reduce the influence of human factors. Based on the BG method, Ma et al. [16] divided 285 cities in China into five types: low, medium-low, medium, medium-high and high. Kamaragugedda et al. [17] divided 15 major cities in Southeast Asia into three types: low, medium, and high. Jiang et al. [4] divided 56 countries in Africa into three types: low, medium, and high. The above researchers used the BG method to divide the individual area (single city or single country). The BG method may cause the same NTL value in different individual areas to be divided into different BG types.

It was necessary for this paper to adopt a unified standard for partitioning to study the spatiotemporal dynamic changes and comparative analysis of different nighttime lighting areas in the transition zone. Therefore, we improved the BG method according to the research purpose and actual situation. By adjusting the split points of the BG, the transition zone lighting area was divided into four sub-regions using a unified standard for partitioning. The approach was as follows: Based upon a graph of the fitted quadratic polynomial (Figure 4), the lighting area was subdivided into four types—low [NTL_0_, NTL_1_), medium [NTL_1_, NTL_2_), high [NTL_2_, NTL_3_), and very high [NTL_3_, NTL_4_]. According to the sampling analysis, low-type BG values were found in rural areas with less human activity; medium-type BG values were found in urban–rural transition areas with more human activity; high-type BG values were found in urban areas with more human activity; and extremely high-type BG values were found in urban central areas with strong human activity.

As shown in Figure 4, the split point P_2_ is the turning point of the parabola, corresponding to the maximum BG value and the medium NTL value; P_0_ and P_4_ are the starting and ending points of the parabola, respectively, corresponding to the minimum NTL_0_ and maximum NTL_4_; P_1_ is the half-point between P_0_ and P_2_ in the vertical coordinate; P_3_ is the half-point between P_2_ and P_4_ in the vertical coordinate.

The calculation of split points is shown in Table 2.

### 3.4. The Relationship between the Classification of Nighttime Lighting Types and Urbanization

Taking Beijing, the capital of China, as an example, the spatial distribution of the population density and land use data in 2020 (Figure 5a,b) were compared with classification results of the nighttime light in the same period (Figure 5c–e).

Through a comparison of Figure 5, it can be seen that the spatial distribution area of low population density (less than 250 people/km^2^) and forest, grassland, and water corresponded to the low-value area of the NTL, BG, and low-type area of nighttime light. The spatial distribution area of medium population density (250–2500 people/km^2^) and crisscross area of farmland and construction land corresponded to the NTL medium-value area, BG high-value area, and medium-type area of nighttime light. The spatial distribution area of high population density (higher than 2500 people/km^2^) and construction land corresponded to the NTL high-value area, BG low-value area, and high and extremely high-type area of nighttime light. A quantitative comparison showed that the sum of high and extremely high NTL areas accounted for 22.84% of the Beijing area, close to the artificial surface areas (21.61%) derived from the land use dataset.

## 4. Results

### 4.1. Long-Term Temporal Trends in Different Nighttime Lighting Types

According to the quadratic relationship between the NTL values and its corresponding BG values at the pixel level of the transition zone, the nighttime light types were divided into four types—low, medium, high, and extremely high—and the long-term temporal trends in different nighttime lighting types from 1984 to 2020 were analyzed (Figure 6).

Figure 6 shows the long-term temporal trends in different types of nighttime lighting areas from 1984 to 2020. The area of each nighttime lighting type in the transition zone displayed an increasing trend, and the increase rate changed from fast to slow according to the low, medium, high, and extremely high types in the past 37 years. The area of low-type NTL increased the fastest, which was 6.66 × 10^3^ km^2^a^–1^. The second was the increase rate of the medium-type NTL area, which was 1.34 × 10^3^ km^2^a^–1^. The increase rate of the high-type NTL area was 0.42 × 10^3^ km^2^a^–1^. The area of extremely high-type NTL increased the slowest, which was 0.19 × 10^3^ km^2^a^–1^. The total area of NTL increased from 1.34 × 10^5^ km^2^ in 1984 to 4.53 × 10^5^ km^2^ in 2020, with an increase of 3.38 times in 37 years, and the average annual growth rate of the NTL area was 3.44%.

### 4.2. Long-Term Spatiotemporal Trends of Nighttime Light at Different Aspects

Taking the county-level administrative region as the unit, the nighttime light in the transition zone was analyzed from the three aspects of area, density, and type.

#### 4.2.1. Spatiotemporal Trend of the Proportion with Nighttime Lighting Area

The area of each county is different. In order to make the results more comparative, the nighttime lighting area (NTLA) proportions (the proportion of NTLA to the total area in each county) were used to analyze and calculate 426 counties in the transition zone. The 400 m contour divided the transition zone into mountainous areas higher than 400 m and plain areas lower than 400 m (Figure 7).

Figure 7 shows the nighttime lighting area proportions. (1) In 1984, the urban distribution was limited by the terrain, with obvious zonality in the Taihang Mountain section. (2) In 2020, urbanization in the Greater Khingan Mountains, Yanshan Mountains, Taihang Mountains, and Xuefeng Mountains sections featured zonality due to topographic constraints. (3) Over the past 37 years, the urban expansion rate broke through the zonal restrictions, especially in the Yanshan and Taihang Mountains.

The specific performance was as follows: In 1984, there were 255 counties with a NTLA proportion that accounted for less than 15%, which was widely distributed in the mountainous areas in the north, south, and middle of the transition zone. There were 171 counties with a NTLA proportion that accounted for more than 15%, which was mainly distributed in the middle plain areas of the transition zone (Figure 7a). In 2020, the number of counties with a NTLA proportion that accounted for less than 15% decreased significantly, with only 62 counties distributed in the northern and some southern mountainous areas of the transition zone. There were 177 counties with a NTLA proportion that accounted for (15%, 60%], which was mainly distributed in the middle mountains and some northern and southern plain areas of the transition zone. There were 187 counties with a NTLA proportion that accounted for more than 60%, which was mainly distributed in the middle plain of the transition zone (Figure 7b). From 1984 to 2020, in the transition zone, there were 112 counties whose growth rate in the NTLA proportion accounted for less than 0.2%, mainly distributed in the northern mountains and middle plains. There were 209 counties whose growth rate of the NTLA proportion accounted for (0.2%, 1.0%], mainly distributed in the northern plain and southern mountainous areas. There were 105 counties with growth rates of a NTLA proportion between 1.0% and 3.0%, which was mainly distributed in the middle plain and the southern plain (Figure 7c). To sum up, from 1984 to 2020, in the transition zone, the NTLA proportion in the plain area was greater than that in the mountainous area, and the NTLA proportion in the middle area was greater than that in the north and south. Among them, the NTLA proportion in the middle plain area tended to be saturated, and the growth rate of NTLA was slow.

#### 4.2.2. Spatiotemporal Trend of the Nighttime Lighting Density

The nighttime lighting density (NTLD) in each county was obtained by the total value of NTL divided by the total area, that is, the NTL value per square kilometer. The NTLD that can reflect the development level of urbanization was analyzed and calculated (Figure 8).

Figure 8a shows the nighttime lighting densities in 1984. The number of counties with a NTLD in the range of [0, 10] was the largest, with 391, and those counties were widely distributed in areas other than large cities. There were 35 counties with a NTLD of (10, 60], which was mainly distributed in provincial capital cities such as Beijing, Shijiazhuang, and Zhengzhou and coal cities such as Fuxin, Yangquan, and Jincheng.

It can be seen from Figure 8b that, in 2020, there were 303 counties with a NTLD of [0, 10], which was mainly distributed in the north, south, and middle of the transition zone. There were 84 counties with a NTLD of (10, 15], which was mainly distributed in the middle plain area, and 39 counties with a NTLD of (15, 60], mainly in Beijing, Shijiazhuang, Zhengzhou, and other provincial capital cities.

According to the slope of nighttime lighting densities from 1984 to 2020 in Figure 8c, the areas with the slowest NTLD growth rate of [0, 0.15] were mainly distributed in the northern and southern mountainous areas, with 129. The number of counties with a NTLD growth rate of (0.15, 0.25] was the largest, with 199, mainly distributed in the middle and southern plain areas. There were 98 counties where the NTLD growth rate was (0.25, 1.00], mainly distributed in the municipal administrative region of the middle and southern plains.

To sum up, from 1984 to 2020, the NTLD in the middle plain of the transition zone was higher, and its growth rate was faster than that in other regions, among which the most obvious was in the capital cities such as Beijing, Shijiazhuang, and Zhengzhou.

#### 4.2.3. Spatiotemporal Trends of the Nighttime Lighting Types

In order to understand the different urbanization types in the transition zone, the NTLA slopes of the four types in 37 years were analyzed and calculated (Figure 9).

Figure 9a shows that the low-type NTLA displayed a downward or constant trend in 111 counties, mainly distributed in the middle plain of the transition zone, with a change rate of (–25, 0] km^2^a^–1^. There were 116 counties with a low-type NTLA slope of (0, 12] km^2^a^–1^, mainly distributed in the southern mountainous area. The low-type NTLA growth rate in the remaining 199 counties was (12, 142] km^2^a^–1^.

In Figure 9b, the middle-type NTLA mainly increased at the rate of (0, 2] km^2^a^–1^ and (2, 5] km^2^a^–1^, with 182 and 140 counties, respectively, which was widely distributed across the whole transition zone. There were 75 counties with a medium-type NTLA that increased at the rate of (5,23] km^2^a^–1^, mainly distributed in the middle and southern plains.

According to the slope changes in the high-type NTLA in Figure 9c, the high-type NTLA displayed a downward or constant trend in 165 counties, mainly distributed in the south of the transition zone. There were 139 counties with a NTLA rate that increased by (0, 1] km^2^a^–1^, mainly distributed in the north, middle, and southern plains. There were 60 counties that increased at a NTLA rate of (1, 10] km^2^a^−1^, mainly distributed in the middle and southern plains.

It can be seen from Figure 9d that the extremely high-type NTLA showed a downward or constant trend in 357 counties. In the remaining areas, the extremely high-type NTLA showed an increasing trend, mainly distributed in the plains. The largest increase in NTLA was in the provincial capital cities such as Beijing, Shijiazhuang, Zhengzhou, and Nanning, and the increase rate reached (2, 26] km^2^a^–1^.

To sum up, the increasing areas of the medium-type NTLA were almost distributed in the whole transition zone. The low-type NTLA showed an increasing trend in areas other than the middle plains. The increased area of the high- and extremely high-type NTLA was mainly distributed in the middle plain.

### 4.3. Spatiotemporal Transitions of Different Nighttime Lighting Types

The variations of nighttime lighting type (NTLT) in different periods can reflect the urbanization process of human activity. Different NTLTs of the whole transition zone and different typical cities in the initial (1984), middle (2002), and end (2020) of the 37 years were selected for spatiotemporal change detection. The different typical cities included first-tier cities (Beijing), new first-tier cities (Zhengzhou), second-tier cities (Shijiazhuang and Nanning), third-tier cities (Guilin and Xinxiang), fourth-tier cities (Shiyan and Huludao), and fifth-tier cities (Zhangjiajie, Fuxin, Xing’an League, and Da Hinggan Ling Prefecture). Sankey diagrams of NTLT transitions (Figure 10) were made to express the proportions and mutual transitions of four NTLTs at three time points: 1984, 2002, and 2020. 

The spatial change detection diagrams of the NTLT (Figure 11) were used to express the spatial distribution of NTL expansion and type transitions in the three periods, which were helpful to understand the dynamic characteristics of the cities in the study area fully.

According to the spatiotemporal transitions of different nighttime lighting types in Figure 10 and Figure 11, during the past 37 years, in the transition zone, a large number of areas without NTL transitioned into areas with NTL, and the higher types of NTL were inherited from the same type in the past or transitioned from lower types. The specific performances were as follows: The NTLA proportion was only 9.64% in 1984. Meanwhile, 12.69% and 0.05% of the study area changed from no NTL to low- and medium-type NTLA, respectively, from 1984 to 2002, and 13.21% and 0.15% of the study area changed from no NTL to low- and medium-type NTLA, respectively, from 2002 to 2020. The large-scale expansion areas of NTL were mainly distributed in the south of the transition zone. During the past 37 years, 4.03% of the low-type NTL remained unchanged, and 3.51% of the low-type NTL transitioned into the higher type, and the transition areas were mainly distributed in the middle plains. By 2020, the proportion of NTLA increased to 34.37%, of which the low, medium, high, and extremely high types accounted for 27.88%, 4.73%, 1.09%, and 0.67%, respectively.

Since the 21st century, China’s economy has developed rapidly, and the process of urbanization has been remarkable, but the changes in the NTLT of different cities have been different. (1) As a first-tier city in China, compared to other cities, Beijing’s extremely high NTLA accounted for the largest and fastest growth rate, with an increase of 12.96% from 1984 to 2020 and 16.53% by 2020 (Figure 10a and Figure 11a). (2) Zhengzhou is a new first-tier city, with the fastest urbanization development. The NTLA transition from the low type to the higher types accounted for the largest (36.31%). The total NTLA of the medium, high, and extremely high types increased from 12.02% in 1984 and 23.11% in 2002 to 50.36% in 2020 (Figure 10b and Figure 11b). (3) As second-tier cities, Shijiazhuang and Nanning had obvious differences in urbanization development due to their different geographical location and population density. Shijiazhuang is located in the Beijing–Tianjin–Hebei Economic Circle of the middle plains, with rapid urbanization. In the 37 years, the NTLA transition from the low type to the higher types reached 18.30% (Figure 10c and Figure 11c). As the capital city of Guangxi, Nanning’s urbanization was relatively backward in the initial period, while the city expanded rapidly in the later period. In 37 years, the urban expansion area reached 51.73% (Figure 10d and Figure 11d). (4) Guilin and Xinxiang are third-tier cities in China. During the past 37 years, the NTLA of Guilin, located in the southern region, increased by 26.02%, and the NTLA of the low, medium, high, and extremely high types increased by 22.40%, 2.84%, 0.75%, and 0.03%, respectively (Figure 10e and Figure 11e). Xinxiang, located in the middle plains, had an unlighted area of only 7.63% in 2020. The NTLA expanded by 50.94% from 1984 to 2002, and 16.46% of the low-type NTLA transitioned to a higher type from 2002 to 2020 (Figure 10f and Figure 11f). (5) The urbanization development characteristics of the fourth- and fifth-tier cities were similar to those of the above cities; the source of each nighttime lighting type mainly came from the same nighttime lighting types and the darker types (Figure 11g–l).

To sum up, the NTLT transitions presented obvious spatiotemporal development modes in the study area. Inland cities generally showed a circular expansion mode from the central area where the high and extremely high types were located to the surrounding areas. Specifically, the extremely high type of nighttime lights remained unchanged, while the medium and high types transitioned into the extremely high type of the city center, and the no and low types transitioned into the medium and high types in the periphery of the city center. The urban expansion of coastal cities, such as Huludao, was limited by the coastline (Figure 11h).

## 5. Relationship between the Spatiotemporal Transitions of Different Nighttime Lighting Types and Terrain

The relationship between urban dynamic characteristics and terrain can be understood by the spatial superposition of DEM and nighttime light data (Figure 12), as well as the statistics of the pixels in the study area and typical cities under different terrain (Appendix A).

It can be seen from Appendix A and Figure 12 that the increase in NTLA in the plains (15.71%) was much higher than that in the mountainous areas (9.09%) over the 37 years. The expansion of nighttime lights mainly extended outward from the boundary of the central area. The NTL in the plains showed a multi-county contiguous expansion pattern (Figure 12d,f,j), and the NTL expansions in the mountainous areas were mostly restricted by the terrain (Figure 12e,g,i). The NTLA in plain and mountainous areas transitioned from the low type to the higher types, respectively accounting for 15.71% and 9.08% of the total areas, which are mainly distributed near the edge of the city or county-level central area, with a contiguous spread of low-type NTL in counties belonging to first- and second-tier cities (Figure 12a–c). The NTLT transitions between the medium, high, and extremely high types mainly occurred in areas close to the city center (Figure 12a–d,f), with 1.14% and 0.16% of the total area converted in the plains and mountains, respectively.

From the transition statistics of the NTLT in each city (Appendix A), it could be found that cities with more than 50% of plains (Figure 12a–d,f,h,j) outperformed cities with more than a 50% mountainous area (Figure 12e,g,i) in terms of nighttime light expansion and type of transitions. In cities spanning the 400 m contour, topographic variations in the plains and mountains hindered large-scale urban expansions and type transitions, such as Beijing, Shijiazhuang, Guilin, Xinxiang, and Zhangjiajie, where urban expansion was significantly limited by elevation changes.

The transition zone was divided into five elevation intervals (<400, 400–800, 800–1200, 1200–1600, and >1600 m) and five slope intervals (<0.5°, 0.5°–1.0°, 1.0°–1.5°, 1.5°–2.0°, and >2.0°) [36], and were analyzed by the correlation between different NTLTs, topographies, and slopes in 1984, 2002, and 2020 (Figure 13).

According to the effects of elevations on the four types of NTLA (Figure 13a–d), the low-type NTLA increased slowly between 2002 and 2020, mainly due to the gradual saturation of the nighttime lighting expandable area in the plains. Elevation had little effect on the distribution and expansion of low-type NTLA (Figure 13a). The medium, high, and extremely high types of NTL were mainly distributed in plains with an elevation of less than 400 m, and the NTLA showed a rapid increasing trend from 2002 to 2020.

Figure 13a–d show the effects of slopes on the four types of NTLA. The four types of NTLA had the largest percentages in areas with a slope of less than 0.5°. The low-type NTLA increased at a faster rate between 1984 and 2002, while the medium-, high-, and extremely high-type NTL increased rapidly between 2002 and 2020. The effects of slope on the increase in nighttime lighting area for the medium and high types were less than those for the low and extremely high types.

Through superposition analysis, it was found that the elevation changes of “plain–mountain” with an altitude of 400 m as the boundary in the transition zone had greater impacts on the urbanization process than the slope.

## 6. Discussion

### 6.1. Comparison of Different Brightness Gradient Methods

According to the research purpose and the actual situation of the research object, we made methodological innovations on the basis of the existing BG method for nighttime lighting imagery. Compared to the existing research methods [4,16,17], there were mainly two differences, as follows.

In order to make the urbanization dynamic characteristics of different cities in the transition zone comparable, we treated the study area as a whole and adopted a unified classification standard to classify its nighttime lighting types. Unlike previous researchers who fitted separate independent quadratic relationships for NTL and BG on local areas (individual cities or individual countries) within the study area [4,16,17], the unified criteria of classification used in this paper can avoid the problem of the same nighttime lighting values in different areas being classified into different nighttime lighting types and can increase the comparability of nighttime light types in different cities.

The transition zone includes multiple city types at the same time (first-tier, new first-tier, second-tier, third-tier, fourth-tier, and fifth-tier). In order to distinguish megacities (first-tier and new-tier) with high nighttime lighting values from general cities better, and to reduce the fragmentation of general urban areas with low nighttime lighting values, the division points of the quadratic relationship curve between NTL and BG were adjusted, and the nighttime lighting types in the transition zone were divided into four types: low, medium, high, and extremely high. Ma et al. [16] divided the nighttime light data of 274 prefecture-level cities in China into five levels: low, medium–low, medium, medium–high, and high. Jiang et al. [4] divided the nighttime light data of 10 major African countries into three types: low, medium, and high. The thresholding split points used in this paper were different from those of previous researchers.

The improved BG method not only enabled the cities in the transition zone to adopt a unified classification standard but also provided a targeted division of nighttime lighting types in cities with different development levels, which can provide ideas for the rapid realization of large-scale dynamic analysis of urban nighttime lighting in long time series.

### 6.2. Urbanization in the Transition Zone

Since the reform and opening up, China’s rapid economic growth has promoted urbanization with rapid development [37]. In the early stage of reform and opening up, due to the influences of the geographical environment, resource industry structure, and national and regional policies, the economy of the eastern plains developed rapidly, while the economy of the central mountains was relatively backward [38,39]. At present, with the national emphasis on coordinated regional economic development, there was an urgent need to understand the dynamics of urbanization in the key region (China’s “plain–mountain transition zone”) where the economy was advancing from east to west [40,41]. As a relatively complete basic geographic unit, the urbanization development of China’s “plain–mountain transition zone” was conducive to optimizing the spatial pattern of regional development, promoting the gradual expansion of economic growth space from east to west, and promoting the coordinated development of east–middle–west, which was of great significance to connect the east and enlighten the west [42].

The research results on the dynamic characteristics of urbanization in the transition zone showed that the nighttime lighting areas of various types experienced an increasing trend, and the nighttime lighting areas of the low type increased the fastest. This conclusion was consistent with the conclusion of Ma et al. [16] on the nighttime lighting type analysis of 273 cities in China. Beijing’s urbanization process was the fastest, and the expansion of core cities such as the Beijing–Tianjin–Hebei Urban Agglomeration and Central Plains Urban Agglomeration dominated the urbanization process [20,43]. The night lighting type transitions of inland cities in the “transition zone” presented a circular expansion mode from the central area where the high type was located to the surrounding lower-type areas, and the urban expansion of coastal cities was affected by the coastline, which is nearly consistent with the urbanization mode of Africa, Southeast Asia, and other regions [4,17].

In addition, the research in the transition zone also found that: (1) The urbanization level decreased with the increase in terrain and the decrease in precipitation. The middle plains are most suitable for the development of urbanization in terms of terrain, precipitation, and other natural factors. The urban expansion speed in the middle of the study area broke through the topographic constraints of the Yanshan and Taihang Mountains (Figure 7c). Although the southern region has sufficient precipitation, the development of urbanization is limited by the terrain of the fragmented distribution of the plain. Due to insufficient precipitation, the urbanization process in the northern region is slow, and the urbanization in the northern mountains is the slowest due to the joint constraints of topography and precipitation. (2) The transition zone (especially the middle plains) is undergoing rapid urbanization. From 1984 to 2020, the nighttime lighting areas expanded rapidly, with an average annual increase of 6821.13 km^2^, and the nighttime lighting areas increased by 3.38 times in the 37 years. Due to the rapid development of urbanization in the middle plains, the expansion of the low-type nighttime lighting area tends to be saturated, and there are potential risks of unsustainable urban expansion. (3) The urbanized area, population, and GDP in the transition zone all showed a significant increasing trend, and the increase in the urbanized area is significantly and highly positively correlated with the increase in population and GDP, respectively (Figure 14).

Rapid urbanization not only brings many negative impacts on the ecological environment and natural resources, resulting in fragmentation of the ecological landscape [44], the reduction in biodiversity, and the increase of greenhouse gas emissions [45,46], but also has a negative impact on urban life, resulting in unemployment [47], an insufficient water supply [48], lack of medical and health infrastructure, and other problems. Therefore, the sustainable development of urbanization is a major problem currently, which requires macro-control of national policies [49]. In the future, urbanization of the transition zone will be further analyzed from the aspects of the population and economy.

### 6.3. Light Pollution

As urbanization accelerates, the scope and intensity of artificial nighttime lighting increases rapidly, and the problem of light pollution ensues. Nighttime lighting may have serious physiological consequences for humans and ecological and evolutionary implications for animal and plant populations, and may reshape entire ecosystems [50]. Therefore, the problem of light pollution has attracted more and more attention. The International Dark-Sky Association (IDA; www.darksky.org (accessed on 6 July 2022)) and the Illuminating Engineering Society of North America (IESNA 2000) have provided preliminary recommendations for lighting regulations and lighting standards. Japan, France, Italy, the United States, and other countries that started light pollution management practice earlier have accumulated useful experience on the issue of light pollution from legal norms, environmental standards, treatment methods, etc. [51].

Light pollution control in China is still in its infancy. China has developed national, industrial, and local standards for light pollution prevention and control in outdoor lighting, advertising, and signage lighting. In outdoor lighting, the Outdoor Lighting Interference Light Restriction Specification (GB/T 35626-2017) specifies the urban environment brightness zoning, interference light classification, and interference light restriction requirements and measures. The Design Code for Urban Night Lighting (JGJ/T 163-2008) gives the light pollution control requirements for urban nighttime lighting. In terms of advertising and signage lighting, the Technical Specification for Urban Outdoor Advertising Facilities (CJJ 149-2010) makes specific provisions for the maximum allowable brightness of outdoor advertising facility lighting. The LED Display Interference Light Evaluation Requirements (GB/T 36101-2018) limit the interference and hazards of reflected light from glass curtain walls and provide the brightness, illumination, and threshold increment limits that should be met by LED displays in different environments.

In addition, China has proposed locally appropriate lighting policies in terms of enhancing people’s environmental awareness of light pollution, developing new technologies, and formulating measures to mitigate light pollution problems [52], such as: (1) Encourage people to turn off unnecessary lighting fixtures and direct light carefully to reduce light pollution by changing the way they think and behave; (2) support the development, promotion, and application of new technologies such as energy-saving lighting sources such as light-emitting diodes, infrared detectors for car lights, and light sensors for street lights; (3) reduce nighttime transportation of freights, which can be shifted to rail or sea transportation.

However, China currently lacks laws and regulations for light pollution, light pollution prevention, and control standards to be improved and implemented effectively. There is an urgent need to combine foreign experience and the actual situation in China for light pollution control [53], which will help countries, regions, and cities to maximize the social and economic benefits of artificial light at night, while minimizing its negative and unintended ecological and health impacts; this will become of great significance to the sustainable development of urbanization.

## 7. Conclusions

Nighttime light data are closely related to human production and life and have become one of the important indicators for studying urbanization. We innovatively used the improved BG method to divided China’s “plain–mountain transition zone” into four nighttime lighting areas: low, medium, high, and extremely high based on the PANDA nighttime light data with the longest coverage time and the latest update time.

It was found that the transition zone has experienced rapid and diverse urbanization processes in the past 37 years. The central region (Yanshan Mountains and Taihang Mountains) had the fastest urbanization development; the northern region (Greater Khingan Mountains) had the slowest urbanization development; and the southern region (Wushan Mountains and Xuefeng Mountains) had medium urbanization development. Whether in the north, middle, or south of the transition zone, the urbanization development of the plains was greater than that of the mountains, with the central region showing the most pronounced performance, and its urban expansion rate broke through the topographic limitation of the plain–mountain areas. However, the rapid development of cities in the middle plains resulted in the expansion of nighttime lights becoming saturated, which has caused an unsustainable potential crisis in urban development in this area. The urbanization of China’s “plain–mountain transition zone” was mainly centered on the extremely high- or high-type nighttime lights, and developed from the extremely high or high type to the surrounding areas in a layer-by-layer annular diffusion pattern.

The unique east–west distribution characteristics of China’s transition zone make the urban expansion and development on both sides of this zone seriously uneven. Although the gap between the east and the west tends to narrow with the development of urbanization, there is still an urgent need for national policies to regulate the coordinated development of urbanization in the transition zone. The results of this study can provide the dynamic characteristics of urbanization in the transition zone, as well as data support for the formulation of sustainable urbanization policies. The problem of light pollution caused by rapid urbanization also needs to be effectively improved through the development of national laws and standards for prevention and control.

## Figures and Tables

**Figure 1 ijerph-19-09230-f001:**
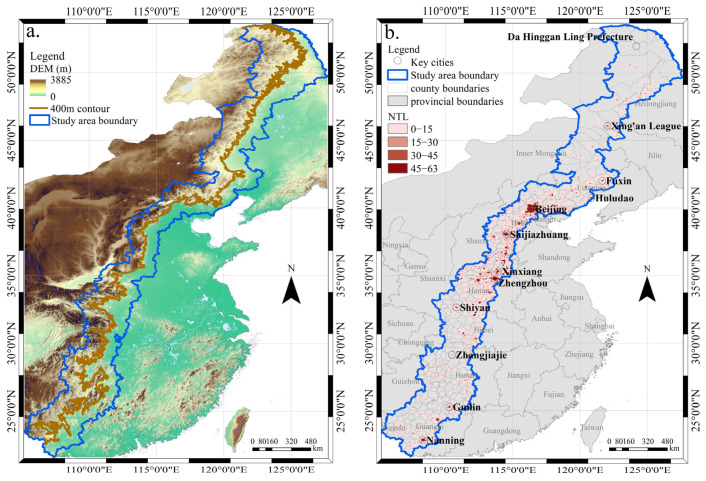
Overview of China’s “plain–mountain transition zone”: (**a**) Topographic map; (**b**) distribution map of nighttime light in 2020.

**Figure 2 ijerph-19-09230-f002:**
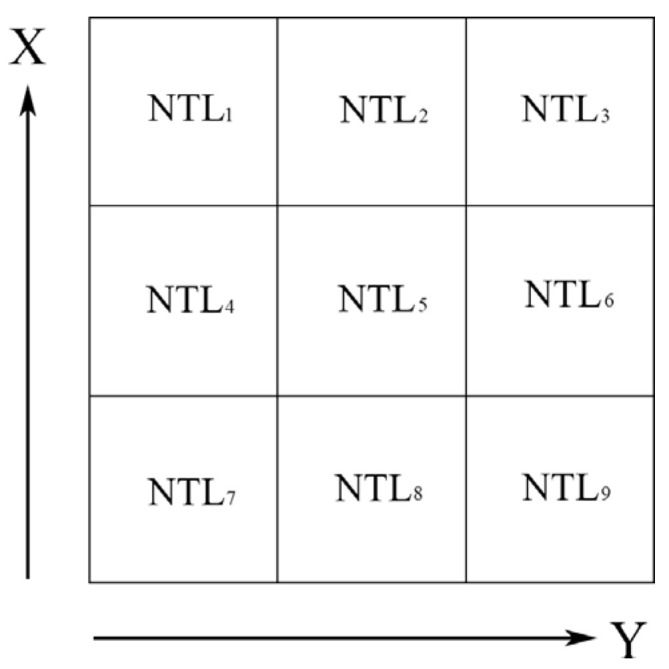
Schematic diagram of the brightness gradients.

**Figure 3 ijerph-19-09230-f003:**
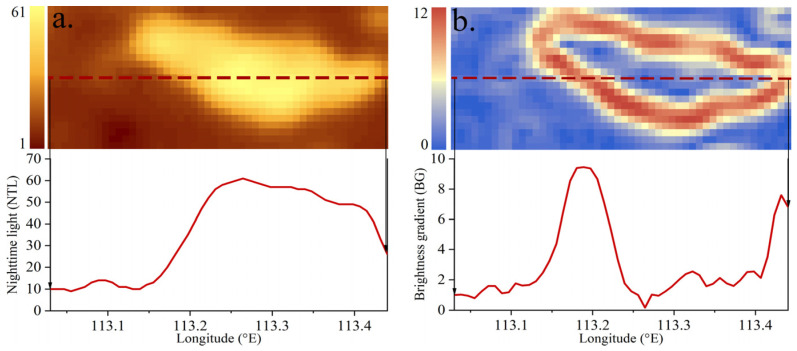
Comparison between nighttime lights and brightness gradients (taking Pingdingshan city of Henan Province as an example): (**a**) Nighttime lights; (**b**) brightness gradients.

**Figure 4 ijerph-19-09230-f004:**
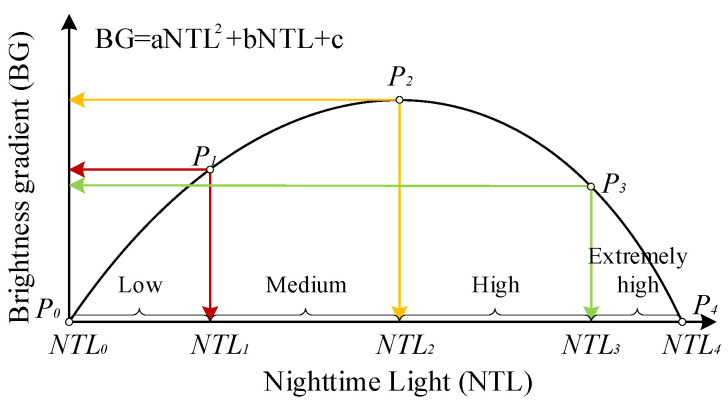
Partition diagram of the quadratic relationship between BG and NTL.

**Figure 5 ijerph-19-09230-f005:**
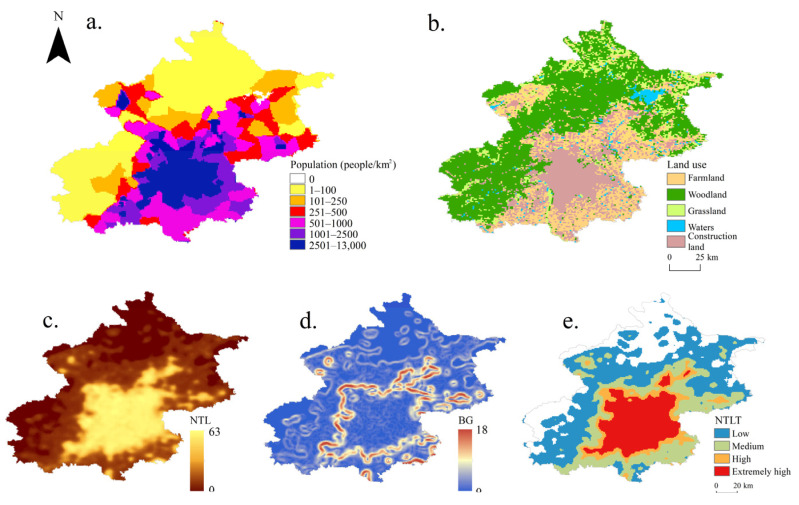
Comparison and verification diagrams of different data types in Beijing, 2020: (**a**) Population data; (**b**) land use data; (**c**) nighttime lights; (**d**) brightness gradients; (**e**) the types of PANDA.

**Figure 6 ijerph-19-09230-f006:**
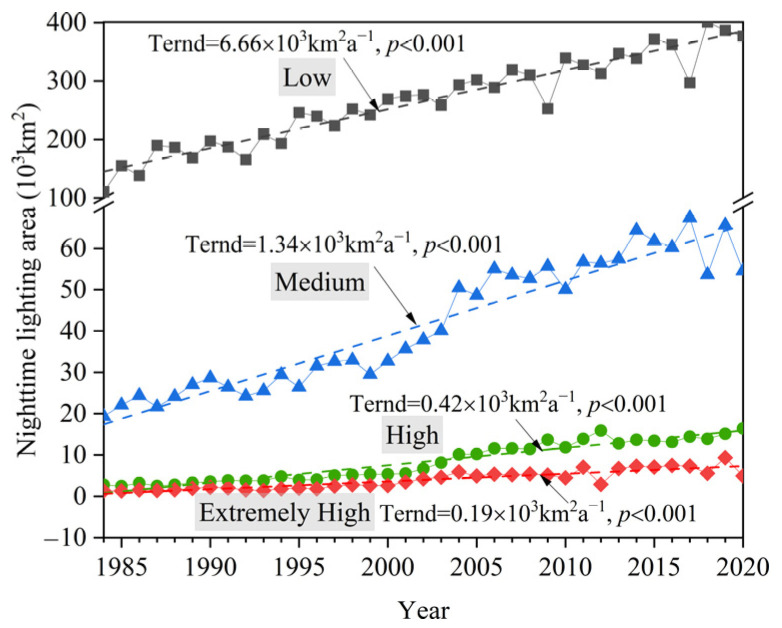
Long-term temporal trends in different types of nighttime lighting areas from 1984 to 2020.

**Figure 7 ijerph-19-09230-f007:**
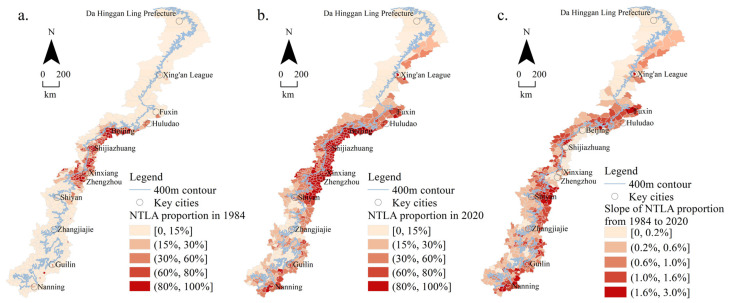
Nighttime lighting area proportions: (**a**) in 1984; (**b**) in 2020; (**c**) slope from 1984 to 2020.

**Figure 8 ijerph-19-09230-f008:**
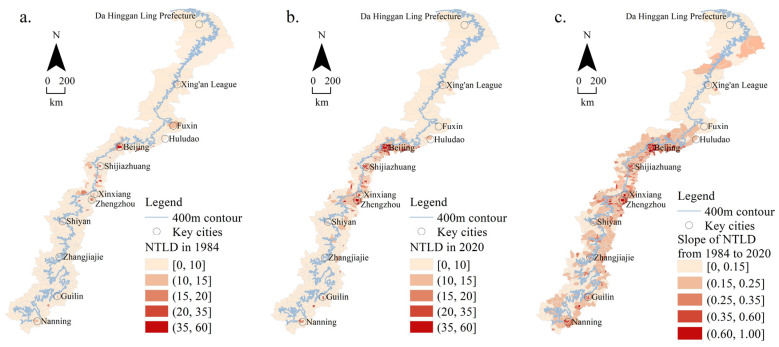
Nighttime lighting densities: (**a**) in 1984; (**b**) in 2020; (**c**) slope from 1984 to 2020.

**Figure 9 ijerph-19-09230-f009:**
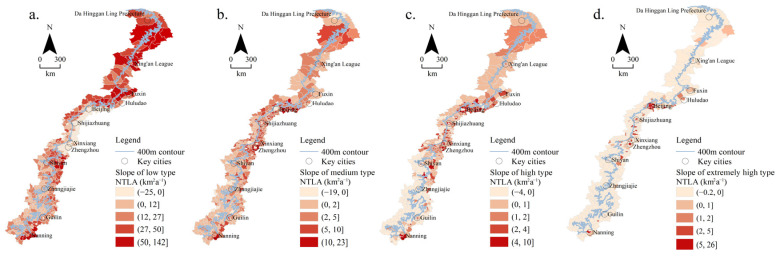
Slope changes in different types of nighttime light: (**a**) low types; (**b**) medium types; (**c**) high types; (**d**) extremely high types.

**Figure 10 ijerph-19-09230-f010:**
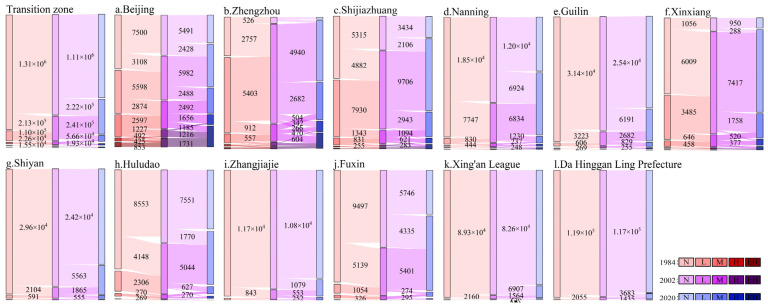
Transitions of different nighttime lighting types in the whole region and different tier cities between 1984, 2002, and 2020. N, none; L, low; M, medium; H, high; EH, extremely high. Notes: The numbers of conversion pixels less than 10,000 in the transition zone and the number of conversion pixels less than 250 in different typical cities were not marked.

**Figure 11 ijerph-19-09230-f011:**
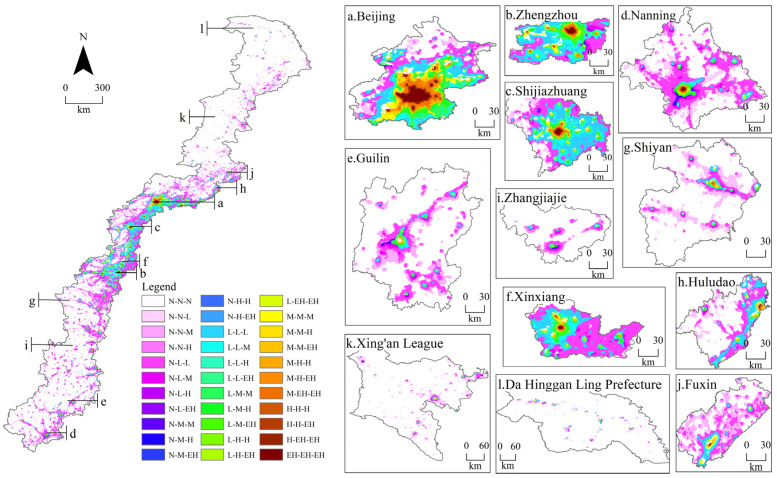
Spatiotemporal transitions of different nighttime lighting types in the whole region and different tier cities between 1984, 2002, and 2020.

**Figure 12 ijerph-19-09230-f012:**
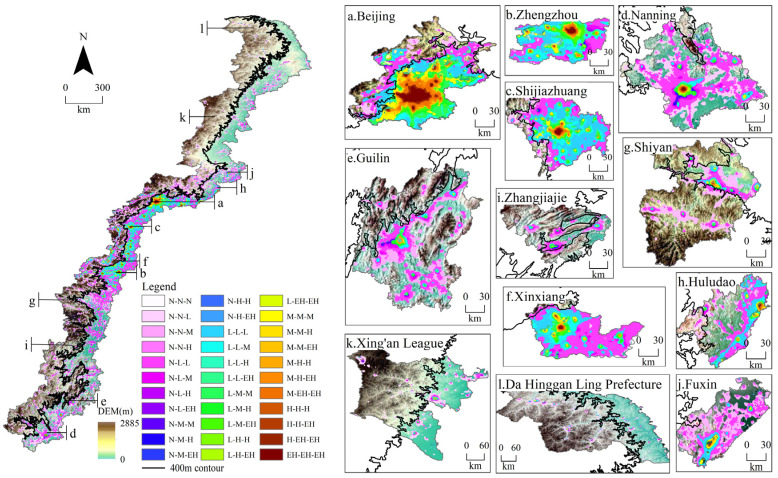
Superposition between the spatiotemporal transitions of different nighttime lighting types and topography.

**Figure 13 ijerph-19-09230-f013:**
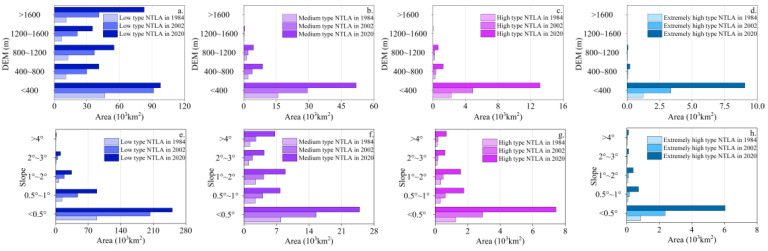
Correlation analysis between the DEM, slope, and different nighttime lighting types in 1984, 2002, and 2020.

**Figure 14 ijerph-19-09230-f014:**
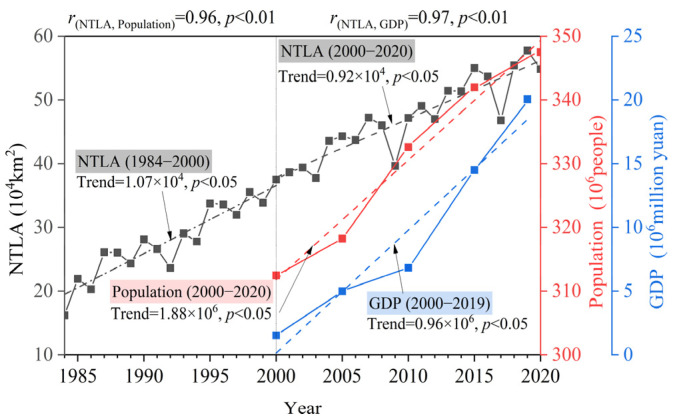
Slope and correlation in NTLA, population, and GDP.

**Table 1 ijerph-19-09230-t001:** The characteristics and use of the data.

Data Type	Time Period	Resolution	Use
PANDA	1984–2020	30″	Research urbanization.
DEM	2000	90 m	Preliminary extraction of the study area boundary and analysis of the response of urbanization to the terrain.
Administrative division	2018	/	Further definition of the study area boundary.
Land use	2020	1 km	Comparison and verification.
Population	2000–2020	30″	Analysis of the relationship between urbanization and population.
GDP	2000–2019	30″	Analysis of the relationship between urbanization and GDP.

**Table 2 ijerph-19-09230-t002:** Split point calculation of quadratic polynomials.

Split Points	Nighttime Light (NTL)	Brightness Gradient (BG)
P_0_ (NTL_0_, BG_0_)	$NTL_{min}$	$aNTL_{min}^{2}+bNTL_{min}+c$
P_1_ (NTL_1_, BG_1_)	$-\frac{b}{2a}-\sqrt{\frac{BG_{1}-c}{a}+\frac{b^{2}}{4a^{2}}}$	$\frac{BG_{0}+BG_{2}}{2}$
P_2_ (NTL_2_, BG_2_)	$-\frac{b}{2a}$	$\frac{-b^{2}+4ac}{4a}$
P_3_ (NTL_3_, BG_3_)	$-\frac{b}{2a}+\sqrt{\frac{BG_{3}-c}{a}+\frac{b^{2}}{4a^{2}}}$	$\frac{BG_{4}+BG_{2}}{2}$
P_4_ (NTL_4_, BG_4_)	$NTL_{max}$	$aNTL_{max}^{2}+bNTL_{max}+c$

## Data Availability

Not applicable.

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
