# Peer review of "Dynamic Characteristics of Urbanization Based on Nighttime Light Data in China’s “Plain–Mountain Transition Zone”"

_ijerph, 2022, doi:10.3390/ijerph19159230_

Round 1

Reviewer 1 Report

The topic is interesting. I think urbanization can be assessed by obtaining the data on light pollution propagation for a long period. It can give us a lot of important conclusions. The abstract and introduction are sufficient. But in my opinion the abstract should consist the main outcome. Moreover, there are some disadvantages that must be improved:

- It is quite hard to catch what was the process of the conducted research. The proper section about it should be added.

- In general, most figures (10,11,12) are low quality. Please provide the figures of very high quality.

- Fig. 1. – It is unclear what the unit of NTL is. Is it in luminance or magnitude per square arc-second? The same comment for the other figures with the brightness scale.

- Please use "(1)" instead of "Eq.1". Please improve the eq. 1. I think the second line is unnecessary because it is also described by eq.2.

- There are many abbreviations in this work, and sometimes, it is hard to catch what is happening. Please explain all abbreviations. Maybe the list of symbols at the end will help?

- What is the main reason for NTL increasing? In general, the answer is that it is the increasing urbanization. However, it depends on many different factors and mainly on outdoor lighting installations. When the installation is not well designed, the light pollution is increased. So, what about the analyzed region's lighting standards and the light pollution reduction policy? How can this be prevented? Some section about it must be added and the therefore, the conclusions should be extended.

Reviewer 2 Report

Because the previous studies on the dynamic characteristics of urbanization mostly focused on individual cities or regions, there was no study on the large-scale geographical transition zone and its response to topography. So this paper intended to select PANDA data with advantages in the time range and data quality, take China’s “plain-mountain transition zone” as the research area, analyze the spatiotemporal characteristics of NTL by improved BG method, and quantitatively evaluate the urban development from 1984 to 2020.

The suggestions are as follows:

(1) Firstly, the abstract is too long, so it needs to be concluded concisely.

(2) The format for paragraph 1 of the Introduction should be modified.

(3) The most important is that what’s the definition of the large-scale geographical transition zone, as well as what’s the characteristics of the large-scale geographical transition zone?

(4) In addition, only the data source is introduced, and the characteristics of the data are ignored. And the data processing is not presented in the manuscript.

(5) Why Sobel operator is used to convolute the nighttime light images and detected the edge of the image? What are the advantages of the Sobel operator?

(6) The English expression is not authentic enough, and the language expression is further polished and modified.

(7) Lastly, the creation points should be presented clearly.

Reviewer 3 Report

This study aims to highlight the dynamic characteristics of urbanization from nighttime light data over China from observed data. The findings are interesting and I have some concerns. Moderate revision is needed to improve the manuscript before publication

1. Please explain what is the north-south differences in urbanization in the "transition zone"?

2. Please indicate the relationship between urbanization (e.g. area and slope) with population and Gross National Product (GDP) (e.g. data and slope, respectively).

Some minor issues:

(1) 'NTLD' should be 'NLD', and 'NLT' should be 'NTL' in the abstract?

(2) 'NTL' should be given the full name in the introduction part.

(3) There are many types of data used in manuscript, and a list should be set up to describe the characteristics of each data (such as type, time period, resolution, etc.) and use.

(4) Please mark the key cities along the "transition zone", in Figure 1b; Figure 7a, b, c; Figure 8a, b, c; Figure 9a, b, c, d;

(5) In Figure 10, the annotation is not clear and needs to be improved.

(6) In Figure 11 and Figure 12, annotations/labels should be in each subfigure, such as "(a) Beijing", etc.

Round 2

Reviewer 1 Report

Dear Authors,

Thank you for the revised version of your paper. I accept all the improvements and answers.

All the best! 

Author Response

Dear editor, Dear reviewer 1,

Thank you again for giving us the opportunity to submit a revised draft of the manuscript “Dynamic Characteristics of Urbanization Based on Nighttime Light Data in Plain-mountain Transition Zone, China” for publication in the Journal of International Journal of Environmental Research and Public Health (IJERPH).

We appreciate the time and effort that you and the reviewer 1 dedicated to providing feedback on our manuscript. Once again, we thank you for the time you put in reviewing our paper and agree to accept our manuscript. Since your inputs have been precious, in the eventuality of a publication, we would like to acknowledge your contribution explicitly.

Sincerely,

Authors.

2022-7-23

Reviewer 2 Report

The manuscript has been modified according to the comments. There are still some questions in the article. The suggestions are as follows:

(1) The language should be polished carefully. Such as, Third, to facilitate area calculation and comparison between different data types, ...

(2) The most important is that the research contents should be presented obviously in the end of induction.

(3) In addition, this article should introduce the research purpose of the research, including the practical and theoretical significance of each contents.

(4) We proposed innovations for the problems existing in data types, research methods and research areas. The innovations should be described in detail.

Reviewer 3 Report

Accept

Author Response

Dear editor, Dear reviewer 3,

Thank you again for giving us the opportunity to submit a revised draft of the manuscript “Dynamic Characteristics of Urbanization Based on Nighttime Light Data in Plain-mountain Transition Zone, China” for publication in the Journal of International Journal of Environmental Research and Public Health (IJERPH).

We appreciate the time and effort that you and the reviewer 3 dedicated to providing feedback on our manuscript. Once again, we thank you for the time you put in reviewing our paper and agree to accept our manuscript. Since your inputs have been precious, in the eventuality of a publication, we would like to acknowledge your contribution explicitly.

Sincerely,

Authors.

2022-7-23